# Source-free Cross-modal Knowledge Transfer by Unleashing the Potential of Task-Irrelevant Data

## Abstract

Source-free cross-modal knowledge transfer is a crucial yet challenging task, which aims to transfer knowledge from one source modality (*e.g.*, RGB) to the target modality (*e.g.*, depth or infrared) *with no access to the task-relevant (TR) source data* due to memory and privacy concerns. A recent attempt Ahmed et al. (2022) leverages the paired task-irrelevant (TI) data and directly matches the features from them to eliminate the modality gap. However, it ignores a pivotal clue that the paired TI data could be utilized to effectively estimate the source data distribution and better facilitate knowledge transfer to the target modality. To this end, we propose a novel yet concise framework to unlock the potential of paired TI data for enhancing source-free cross-modal knowledge transfer. Our work is buttressed by two key technical components. Firstly, to better estimate the source data distribution, we introduce a **T**ask-irrelevant data-**G**uided **M**odality **B**ridging (**TGMB**) module. It translates the target modality data (*e.g.*, infrared) into the source-like RGB images based on paired TI data and the guidance of the available source model to alleviate two key gaps: 1) inter-modality gap between paired TI data; 2) intra-modality gap between TI and TR target data. We then propose a **T**ask-irrelevant data-**G**uided **K**nowledge **T**ransfer (**TGKT**) module that transfers knowledge from the source model to the target model by leveraging paired TI data. Notably, due to the unavailability of labels for the TR target data and its less reliable prediction from the source model, our TGKT model incorporates a self-supervised pseudo-labeling approach to enable the target model to learn from its own predictions. Extensive experiments show that our method achieves the state-of-the-art performance on three datasets (RGB-to-depth and RGB-to-infrared).

## 1 Introduction

In recent years, researchers have extensively utilized depth or infrared sensors to broaden the scope of computer vision applications beyond the use of RGB cameras Hao et al. (2021); Zhang et al. (2022); Zhou et al. (2022); Lin et al. (2022); Munaro et al. (2014); Zhou et al. (2021); Chang et al. (2017). However, learning successful deep learning-based models for depth and infrared modalities necessitates a significant amount of labeled data for supervision. Acquiring large and diverse datasets incurs prohibitively high costs for data annotation. Consequently, research endeavors have been dedicated to exploring the cross-modal distillation methods Sun et al. (2021); Hafner et al. (2018); Wang et al. (2021) that transfer knowledge from a model trained on extensively labeled RGB data to the target modality, such as depth and infrared.

Nevertheless, practical limitations, *e.g.*, memory constraints and privacy concerns, may render these labeled datasets unavailable in real-world scenarios. In light of this, SOCKET Ahmed et al. (2022) presents a pioneering approach to address the challenge of source-free cross-modal knowledge transfer by learning a model for task-relevant (TR) target modality data with only accessing to source model pretrained by TR source modality data. Specifically, it involves: a) a source model trained for the task of interest (*i.e.*, classification); b) unlabeled TR data in the target modality with the same task of interest; and c) paired task-irrelevant (TI) data in both the source and target modalities. Notably, SOCKET proposes an intuitive strategy of directly reducing the distance of features between paired TI data to mitigate the modality gap.

However, we observe that the paired TI data plays a crucial role in bridging the modality gap by effectively estimating the missing source data distribution and facilitating knowledge transfer from the source to the target modalities. Drawing inspiration by the prior source-free domain adaptation (SFDA) methods Li et al. (2020); Liu et al. (2021), we find that the paired TI data can be effectively

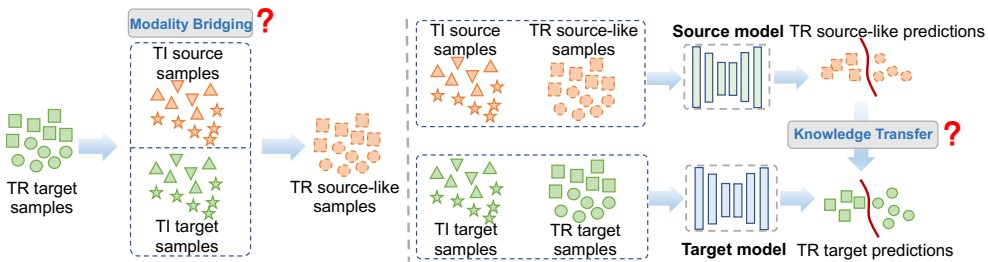

Figure 1: Our observations. Paired TI data facilitates translating TR target data into TR source-like RGB images and transferring knowledge for the task of interest.

utilized to translate the TR target data (*e.g.*, infrared) into the TR source-like RGB data, which aligns well with the original source data distribution, as illustrated in Fig. 1. However, naively applying these SFDA methods without considering the significant modality gap leads to a marginal or even deteriorated performance boost, as confirmed by our experiments in Tabs. 2 and 3. Moreover, the paired TI data (*e.g.*, infrared) can be coupled with TR target data to extract knowledge from the source model and transfer it to the target model. A straightforward way is to exploit cross-modal distillation methods, *e.g.*, Gupta et al. (2016); Garcia et al. (2019); Sayed et al. (2019); Hoffman et al. (2016); however, these methods are limited to transferring knowledge solely with the paired TR data. In this work, there exists the paired TI data, which may have the potential to facilitate knowledge transfer for the task of interest, as shown in Fig. 1. With these observations, we strive to address a novel question: *"how to unlock the potential of the paired TI data to accurately estimate the source data distribution to alleviate the modality gap, and facilitate knowledge transfer from the source model to the target model for the unlabeled TR target data?"*

In this paper, we present a novel yet concise framework comprising two core components: Task-irrelevant data-Guided Modality Bridging module (**TGMB**) and Task-irrelevant data-Guided Knowledge Transfer (**TGKT**) module, as depicted in Fig. 2. Specifically, the TGMB module is introduced to translate the TR target data into the TR source-like RGB images via designing a translation net (See Sec. 3.1). This allows for mitigating the large modality gap between RGB and depth/infrared modalities. Specifically, to generate source-like RGB images with the paired TI data, we employ domain-adversarial learning Creswell et al. (2018) to eliminate two primary gaps: the inter-modality gap between the paired TI data; the intra-modality gap between the TI and TR target data. Note that *using existing image translation methods, e.g., Razavi et al. (2019); Esser et al. (2021), alone does not meet our specific needs as they primarily aim to reconstruct natural-looking images rather than optimal source-like RGB images (i.e., inputs) for the source model.* Therefore, to guide the translation process, we utilize the available source model to maximize the mutual information Ahmed et al. (2021); Liang et al. (2020a) between the distribution of the TR source-like data and its corresponding predictions generated by the source model. This ensures that the translated source-like RGB images closely resemble the source data distribution required for effective knowledge transfer.

After bridging the source and target modalities, we propose a TGKT module that transfers the knowledge of the source model to the target model for the unlabeled target modality (See Sec. 3.2). As the predictions of TR source-like images may be less reliable due to the absence of the ground-truth labels and the modality gap between the source and target data, a technical challenge arises: *"how to effectively transfer knowledge from the source model with less reliable predictions to the target model using the paired TI data?"*. To overcome this challenge, we first transfer knowledge for the task of interest by minimizing the KL-divergence between the predictions of TR source-like RGB images and TR target data. Furthermore, we utilize the paired TI data to facilitate knowledge transfer by decreasing the distance between the features of TI source and target data. Considering the limitations of the TR source-like images' predictions, we incorporate a self-supervised pseudo-labeling approach Liang et al. (2020a) to enable the target model to learn from its own predictions, thereby mitigating the impact of less reliable predictions.

We evaluate the effectiveness of our proposed method on two cross-modal knowledge transfer tasks with three benchmark datasets: SUN RGB-D Song et al. (2015), DIML RGB-D Cho et al. (2021), and RGB-NIR Brown & Süsstrunk (2011). Our proposed method achieves a performance improvement of **+9.81%** on the DIML RGB-D dataset and **+3.50%** on the RGB-NIR dataset, surpassing the state-of-the-art methods.

## 2 RELATED WORK

**Source-Free Cross-Modal Knowledge Transfer.** SOCKET Ahmed et al. (2021) introduces a pioneering approach that addresses the challenging problem of transferring knowledge from one source modality (*e.g.*, RGB) to the target modality (*e.g.*, depth or infrared) with no access to the TR source data. However, SOCKET primarily focuses on reducing the modality gap by directly decreasing the feature distance between the paired TI data. In contrast, we find that the paired TI data can be coupled with TR target data to extract knowledge from the source model and transfer it to the target model. Consequently, we strive to *unlock the potential of the paired TI data to estimate the source data distribution to alleviate the modality gap, and facilitate knowledge transfer from the source model to the target model for unlabeled TR target data.*

**Source-Free Domain Adaptation.** To address data privacy and storage concerns associated with unsupervised domain adaptation (UDA) methods Ben-David et al. (2006); Pan & Yang (2010); Zhu et al. (2023), source-free domain adaptation (SFDA) methods have emerged, which can be broadly categorized into data generation-based approaches Li et al. (2020); Liu et al. (2021) and model fine-tuning-based approaches Liang et al. (2020b; 2021); Ahmed et al. (2021); Ding et al. (2022); Yang et al. (2021). Data generation-based methods adopt the generative model to estimate the source data distribution by generating source-like data to improve the model performance in the target domain. On the other hand, model fine-tuning-based methods exploit information maximization, self-supervised pseudo-label refinement, spherical k-means, and attention mechanism to achieve a single source adaptation to an unlabeled target domain. However, it is worth noting that data generation-based methods may generate noisy source-like images and model fine-tuning-based methods generate noisy pseudo labels for target data. In contrast to the aforementioned approaches, *source-free cross-modal knowledge transfer presents additional challenges due to the need to transfer knowledge between different modalities. Leveraging the availability of the paired TI data and the available source model, our proposed method aims to estimate source-like RGB images to bridge the modality gap.*

**Cross-Modal Distillation**. To transfer knowledge across different modalities, cross-modal knowledge distillation (CMKD) methods are typically used, which try to transfer knowledge learned from a large-scale labeled dataset of one modality to another modality without a large amount of labeled data Gupta et al. (2016). Most existing CMKD methods Gupta et al. (2016); Garcia et al. (2019); Sayed et al. (2019); Hoffman et al. (2016); Ferreri et al. (2021); Du et al. (2019); Ayub & Wagner (2019) highly rely on the assumption that the paired TR data is available across different modalities. However, conventional CMKD methods only focus on transferring knowledge using the paired TR data. In this work, *we strive to utilize the paired TI data to facilitate knowledge transfer for the task of interest in our source-free cross-modal knowledge transfer problem.*

## 3 PROPOSED FRAMEWORK

**Overview** Assume we are given the well-annotated source modality data $\mathcal{D}s = \{(x_i^s, y_i^s)\}_{i=1}^{n_s}$, consisting of $n_s$ labeled samples. Here, $x_i^s \in \mathcal{X}^s$ and $y_i^s \in \mathcal{Y}^s \subseteq \mathbb{R}^K$, where $K$ denotes the total number of classes in the label set $\mathcal{C} = 1, 2, \cdots, K$. Similarly, $\mathcal{D}_t = \{(x_i^t)\}_{i=1}^{n_t}$ represents the target domain dataset, comprising $n_t$ unlabeled samples, which share the same underlying label set $\mathcal{C}$ as the source domain. We define paired TI data as $\{x_{TI_i}^s, x_{TI_i}^t\}_{i=1}^{n_{TI}}$, where $x_{TI_i}^s$ corresponds to the $i$-th TI data point from the source modality, and $x_{TI_i}^t$ is its corresponding counterpart from the target modality. The total number of paired TI data is denoted as $n_{TI}$. In the source-free cross-modal knowledge transfer scenario, we have access to the pre-trained source model $\mathbf{C}(\mathbf{F}_s(\cdot))$, which has been trained on $\mathcal{D}_s$ using supervised learning with a cross-entropy loss. Here, $\mathbf{F}_s$ represents the CNN feature extractor followed by a linear classifier $\mathbf{C}$. Only $\mathcal{D}_t$ is available during the training, and no data in $\mathcal{D}_s$ can be utilized. Additionally, we use $\mathbf{F}_s(x)$ to denote feature representations and introduce the translation net $\mathbf{T}$ to convert single-channel depth/infrared data into three-channel source-like RGB images suitable for the source model. To mitigate the inter-modality and intra-modality gaps, we employ two discriminators, $\mathbf{D}_1$ and $\mathbf{D}_2$, respectively. The objective of our work is to learn a target-specific feature extractor $\mathbf{F}_t$ that generates target representations aligned with the source data representations by leveraging the source feature extractor $\mathbf{F}_s$ and the paired TI data.

The proposed framework is depicted in Fig. 2, consisting of two main components: TGMB and TGKT. The TGMB module aims to translate TR target data into TR source-like RGB images that are compatible with the source model (See Sec. 3.1). The modality bridging is achieved by leveraging

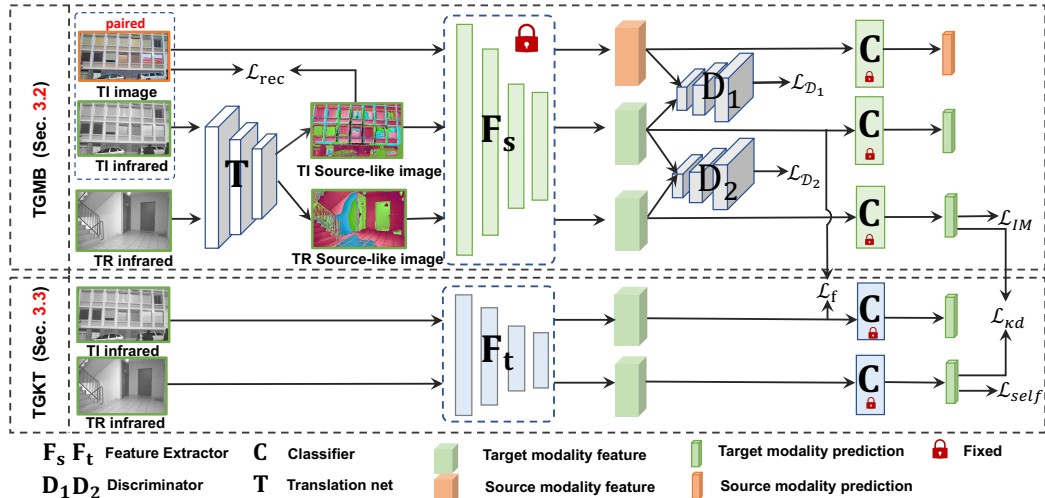

Figure 2: Overall framework of our proposed method. TGMB: Task-irrelevant data-Guided Modality Bridging, TGKT: Task-irrelevant data Guided Knowledge Transfer.

the paired TI data and the pre-trained source model to generate TR source-like RGB data, enabling us to extract source knowledge and train the translation net accordingly. Subsequently, in the TGKT process (See Sec. 3.2), we transfer knowledge from the source model with less reliable predictions to the target model using the paired TI data. This facilitates the transfer of learned knowledge from the source model to the target model, enhancing its performance in the task of interest. The modality gap is effectively eliminated through the synergistic operation of these modules, empowering the target model to acquire the knowledge learned from the source model. We now describe these technical components in detail.

### 3.1 Task-irrelevant data-Guided Modality Bridging (TGMB)

In our source-free cross-modal knowledge transfer setting, we have access to the paired TI data, which can be leveraged to mitigate the modality gap. Unlike SOCKET Ahmed et al. (2022), which directly minimizes the distance between the paired TI data in the feature space to align the distributions of the two modalities, we further propose a novel approach to generate source-like RGB images that effectively bridge the modality gap. However, *there exists a challenging question of how to translate TR target data into TR source-like RGB images while maintaining alignment with the source data distribution.* To tackle this challenge, we introduce the TGMB module, as illustrated in Fig. 2. The TGMB module consists of a translation net $\mathbf{T}$ responsible for converting one-channel depth/infrared data into three-channel RGB images, along with two discriminators $\mathbf{D}_1$ and $\mathbf{D}_2$ designed to minimize both the inter-modality gap and the intra-modality gap, respectively. Moreover, we leverage the knowledge embedded in the pre-trained source model $\mathbf{F}_s$ to guide the training of the translation net by utilizing the mutual information of the TR source-like RGB images.

Specifically, the TGMB module is designed to translate the one-channel target depth/infrared modality data into the source-like RGB data. To accomplish this, we propose a training objective that focuses on reconstructing the TI source RGB images through the translation of TI target data. The objective is formulated as follows:

$$\mathcal{L}_{rec} = \frac{1}{n_{TI}} \sum_{i=1}^{n_{TI}} \left\| x_{TI_i}^s - \mathbf{T}(x_{TI_i}^t) \right\|^2. \tag{1}$$

To facilitate the translation process, our approach centers around reducing two gaps: the inter-modality gap between the paired TI data; 2) the intra-modality gap between the TI and TR target data. The primary objective of minimizing the inter-modality gap is to discriminate between the TI source RGB images and the translated TI source-like RGB images. To ensure the compatibility of the source-like RGB images with the available source model, we employ an adversarial learning procedure incorporating a feature-based (instead of image-based) discriminator $\mathbf{D}_1$. The objective

function for this procedure is formulated as follows:

$$\mathcal{L}_{D_1} = \frac{1}{n_{TI}} \sum_{i=1}^{n_{TI}} \log(\mathbf{D}_1(\mathbf{F}_s(x_{TI_i}^s))) + \frac{1}{n_{TI}} \sum_{i=1}^{n_{TI}} \log(1 - \mathbf{D}_1(\mathbf{F}_s(\mathbf{T}(x_{TI_i}^t)))). \tag{2}$$

Given the absence of explicit supervision for the translation of TR target data into TR source-like RGB images, we address this challenge by focusing on minimizing the intra-modality gap between TI and TR target data. This ensures an effective translation process that successfully aligns the TR target data with the TR source-like RGB images. To achieve this, we introduce the feature-based discriminator $\mathbf{D}_2$ and define the corresponding objective as follows:

$$\mathcal{L}_{D_2} = \frac{1}{n_t} \sum_{i=1}^{n_t} \log(\mathbf{D}_2(\mathbf{F}_s(\mathbf{T}(x_i^t)))) + \frac{1}{n_{TI}} \sum_{i=1}^{n_{TI}} \log(1 - \mathbf{D}_2(\mathbf{F}_s(\mathbf{T}(x_{TI_i}^t)))). \tag{3}$$

The overall objective of the discriminators $\mathbf{D}_1$ and $\mathbf{D}_2$ is to jointly minimize the inter-modality and intra-modality gaps for effective translation and alignment of the TR target data with the TR source data. And the objective of discriminators is formulated as

$$\mathcal{L}_D = \mathcal{L}_{D_1} + \mathcal{L}_{D_2}. \tag{4}$$

After applying reconstruction and discriminator losses following existing image translation methods Razavi et al. (2019); Esser et al. (2021); Van Den Oord et al. (2017), the translation net is capable of generating natural-looking images. However, note that these images may not fully meet our specific requirements and align with the original source data distribution. Therefore, to improve the suitability of the TR source-like images for the source model, we incorporate mutual information Ahmed et al. (2021); Liang et al. (2020a) to guide the translation process. Specifically, we compute the conditional entropy $\mathcal{L}_{ent}$ and the marginal entropy, referred to as diversity $\mathcal{L}_{div}$, for the TR source-like RGB data. These measures are designed to capture the information content and distributional consistency of the TR source-like RGB images, thus facilitating effective translation and bridging of the modality gap. The mutual information loss $\mathcal{L}_{IM}$ is defined as

$$\mathcal{L}_{IM} = \mathcal{L}_{ent} - \mathcal{L}_{div}, \tag{5}$$

where $\mathcal{L}_{ent} = -\frac{1}{n_t} \left[ \sum_{i=1}^{n_t} \delta_k(\mathbf{C}(\mathbf{F}_s(\mathbf{T}(x_i^t)))) \log \delta_k(\mathbf{C}(\mathbf{F}_s(\mathbf{T}(x_i^t)))) \right]$, $\mathcal{L}_{div} = -\sum_{k=1}^{K} \bar{p}_k \log \bar{p}_k$. Here, $\bar{p} = \frac{1}{n_t} \left[ \sum_{i=1}^{n_t} \delta_k(\mathbf{C}(\mathbf{F}_s(\mathbf{T}(x_i^t)))) \right]$ represents the embedding of the entire domain and $\delta_k$ denotes the $k$-th element in the softmax output of the classifier. Finally, the total objective of TGMB is obtained by combining the aforementioned losses, resulting in the following formulation:

$$\mathcal{L}_{TGMB} = \mathcal{L}_{rec} + \alpha_d \mathcal{L}_D + \alpha_{im} \mathcal{L}_{IM}, \tag{6}$$

where the trade-off parameters $\alpha_d$ and $\alpha_{im}$ control the relative importance of each loss term.

## 3.2 TASK-IRRELEVANT DATA-GUIDED KNOWLEDGE TRANSFER (TGKT)

After TGMB module successfully translates TR target data into TR source-like RGB images that conform to the distribution of the source modality data, our approach focuses on leveraging both the paired TR target data and source-like RGB images to transfer knowledge from the source model to the target modality data. In contrast to previous CMKD methods Sun et al. (2021); Hafner et al. (2018); Wang et al. (2021) that solely rely on the paired TR data, our work also takes advantage of the available paired TI data. Moreover, our approach faces a challenge of the absence of ground-truth labels for the target data and the less reliability of predictions made by the source model on the TR source-like images. Consequently, a crucial question arises: *"How to effectively utilize the TI data to facilitate knowledge transfer from the source model with less reliable predictions to the target model for the task of interest?"* In response to this question, we introduce the Task-irrelevant data Guided Knowledge Transfer (TGKT) as shown in Fig. 2.

First, in line with prior CMKD methods, we integrate KL-divergence into our approach to facilitate knowledge transfer from the source model to the target model, leveraging the TR target data and the generated TR source-like RGB data. The objective function for knowledge transfer is formulated as:

$$\mathcal{L}_{kd} = \frac{1}{n_t} \sum_{i=1}^{n_t} KL\left(\delta_k(\mathbf{C}(\mathbf{F}_t(x_i^t)) \| \delta_k(\mathbf{C}(\mathbf{F}_s(\mathbf{T}(x_i^t)))\right). \tag{7}$$

In light of the availability of paired TI data, we propose a knowledge transfer approach between models that capitalizes on this data. Specifically, we facilitate the learning process of the target model by minimizing the distance between paired TI data in the feature space. This strategy aims to align the feature representations of the two modalities, thereby enabling effective knowledge transfer from the source model to the target model. To achieve this, we define the following objective:

$$\mathcal{L}_f = \frac{1}{n_{TI}} \sum_{i=1}^{n_{TI}} \left\| \mathbf{F}_s(x^s_{TI_i}) - \mathbf{F}_t(x^t_{TI_i}) \right\|^2 . \tag{8}$$

However, due to the absence of ground-truth labels for the TR data and the modality gap between the source and target data, the predictions made by the source model on the TR target data are somewhat less accurate. If only KL-divergence and feature matching losses are employed, the performance of the target model will be limited. Therefore, to effectively tackle this issue, we further adopt a self-supervised pseudo-labeling method Liang et al. (2020a) to enable the target model to learn from its own predictions. This approach leverages the inherent structure or properties of the target data to create pseudo-labels, which serve as supervision signals for training the target model. The objective of such an approach is formulated as follows:

$$\mathcal{L}_{self} = \frac{1}{n_t} \sum_{i=1}^{n_t} \mathbf{1}_{[k=\hat{y}_t]} \log \delta_k(\mathbf{C}(\mathbf{F}_t(x^t_i))), \tag{9}$$

where $\mathbf{1}.$ is an indicator function that evaluates to 1 when the argument is true. $\hat{y}_t$ is the pseudo-label of the $i$-th target data, which is obtained via k-means clustering as Liang et al. (2020a). This objective encourages the target model to make accurate predictions, leveraging the TR target data in a self-supervised manner. And the total objective of TGKT is defined as the combination of three key terms:

$$\mathcal{L}_{TGKT} = \mathcal{L}_{kd} + \beta_f \mathcal{L}_f + \beta_{self} \mathcal{L}_{self}, \tag{10}$$

where $\beta_f$ and $\beta_{self}$ are the balancing hyper-parameters.

## 4 EXPERIMENTS

### 4.1 DATASETS, BASELINES, AND IMPLEMENTATIONS

**Datasets.** To testify the versatility of the proposed method, we conduct experiments on the several public visual datasets: SUN RGB-D Song et al. (2015), DIML RGB+D Cho et al. (2021), and RGB-NIR Brown & Süsstrunk (2011). *The details about datasets are shown in the suppl. material.* The detailed statistics of the datasets is illustrated in Tab. 1.

**Baseline methods.** This paper addresses a novel problem statement that has received limited attention in the existing literature, with SOCKET Ahmed et al. (2022) being the only prior work that has considered this specific problem. To assess the effectiveness of our proposed method, we conduct comprehensive comparisons with SHOT Liang et al. (2020a), which is widely recognized as the state-of-the-art approach in the field of SFDA. By selecting SOCKET and SHOT as baseline methods, we aim to provide a rigorous evaluation of our proposed approach against the most relevant and competitive existing techniques.

**Implementations.** In our experimental setup, we implement our proposed method using PyTorch framework. For training the source model, we utilize the widely adopted ResNet50 architecture He et al. (2016) pretrained on the ImageNet dataset Deng et al. (2009), following previous works Liang et al. (2020a); Peng et al. (2019); Xu et al. (2019). To construct the translation net, we combine a fully connected layer, a batch normalization layer, and a convolutional layer. Similarly, for the two discriminators, we employ a three-layer fully connected network architecture. *Due to the page limit, we put the details in the suppl. material.*

### 4.2 EXPERIMENTAL RESULTS

**Results on the SUN RGB-D dataset.** Tab. 2 presents the results obtained on the SUN RGB-D benchmark, demonstrating the superiority of our proposed method over the state-of-the-art approaches in terms of average performance. The proposed TGMB module exhibits a remarkable performance

Table 1: TR/TI split on the three datasets

|  | SUN-RGBD | DIML | RGB-NIR |
|---|---|---|---|
| Number of domains | 4 | 1 | 1 |
| Domain names | K-v1, K-v2, Real, Xtion | N/A | N/A |
| # of TR images for source training | 1258, 2250, 727, 2528 | 693 | 315 |
| # of TR unlabeled images | 1258, 2250, 727, 2528 | 693 | 315 |
| Number of paired TI images | 3572 | 1419 | 162 |
| Number of TR & TI classes | 17&28 | 6&12 | 6&3 |
| Modalities | RBG-D | RGB-D | RGB-NIR |

Table 2: Classification accuracy (%) on the **SUN RGB-D** dataset.

| Target Depth / Source Image | K-v1 | | | | | K-v2 | | | | |
|---|---|---|---|---|---|---|---|---|---|---|
|  | Source only | SHOT | SOCKET | TGMB | TGKT | Source only | SHOT | SOCKET | TGMB | TGKT |
| K-v1 | 22.73 | 16.38 | 29.17 | 34.50 | **41.49** | 18.27 | 17.78 | 19.16 | 10.33 | 16.71 |
| K-v2 | 3.02 | 10.81 | **19.63** | 12.26 | 18.76 | 20.36 | 43.87 | 48.58 | 29.11 | **51.47** |
| Real | 4.45 | 5.33 | **17.09** | 8.74 | 15.66 | 8.09 | 11.29 | 42.67 | 16.80 | **44.49** |
| Xtion | 2.70 | 10.89 | 25.83 | 15.82 | **28.62** | 8.13 | 20.67 | 38.62 | 20.40 | **41.16** |
| Average | 8.23 | 10.85 | 23.60 | 17.76 | **26.13** | 13.71 | 23.40 | 37.26 | 19.16 | **38.46** |

| Target Depth / Source Image | Real | | | | | Xtion | | | | |
|---|---|---|---|---|---|---|---|---|---|---|
|  | Source only | SHOT | SOCKET | TGMB | TGKT | Source only | SHOT | SOCKET | TGMB | TGKT |
| K-v1 | 5.19 | 7.29 | 12.93 | 8.52 | **13.20** | 10.88 | 17.60 | 18.04 | 13.69 | **24.49** |
| K-v2 | 13.76 | 19.81 | 32.46 | 18.98 | **39.20** | 10.84 | 11.04 | 27.69 | 16.81 | **28.14** |
| Real | 11.83 | 15.27 | 44.15 | 28.06 | **46.49** | 6.25 | 8.50 | 23.22 | 14.16 | **24.92** |
| Xtion | 6.19 | 18.43 | **27.65** | 12.28 | 25.58 | 5.81 | 11.04 | **40.70** | 18.30 | 39.20 |
| Average | 9.24 | 15.20 | 29.30 | 19.96 | **31.12** | 8.45 | 12.05 | 27.41 | 15.74 | **29.19** |

Table 3: Classification accuracy (%) on **DIML RGB-D** and **RGB-NIR** datasets.

| Method | Source only | SHOT | SOCKET | TGMB | TGKT |
|---|---|---|---|---|---|
| RGB -> Depth | 26.55 | 39.97 | 40.98 | 44.30 | **50.79** |
| RGB->NIR | 76.83 | 86.03 | 86.98 | 81.59 | **90.48** |

improvement compared to the source-only approach, with gains of **+9.53%**, **+5.45%**, **+10.72%**, and **+7.29%** observed across four domains, respectively. These results indicate the efficacy of our TGMB module in effectively bridging the modality gap by translating TR target data into TR source-like RGB images with the paired TI data and the source model. In the K-v1 → K-v1 transfer task, TGMB even outperforms SOCKET by **+5.33%**, further validating its effectiveness in mitigating modality gaps. Based on the TGMB module, our TGKT method achieves the highest performance in 11 out of 16 transfer tasks and has a gain of **+2.53%**, **+1.20%**, **+1.82%**, and **+1.78%** in four domains over SOCKET. This outcome highlights the superiority of TGKT, as it enables the paired TI data to facilitate knowledge transfer for the TR data, allowing the target model to learn from the source model with less reliable predictions. Specifically, TGKT achieves performance gains of **+12.32%** and **+6.74%** over SOCKET in K-v1 → K-v1 and K-v2 → Real tasks, respectively. However, due to the limitations of the translation process, TGMB achieves a performance of 10.33% compared to the 18.27% achieved by the source-only approach in K-v1 → K-v2 task, resulting in that TGKT achieves a 2.45% performance drop compared to SOCKET. This discrepancy may arise from the utilization of the source model to guide the translation, with the less reliable mutual information about TR data for some tasks, leading to less satisfactory performance of TGMB. *A comprehensive analysis can be found in the suppl. material.* Overall, our proposed method has delivered compelling results, validating its effectiveness in addressing the source-free cross-modal knowledge transfer problem.

**Results on the DIML and RGB-NIR datasets.** To further demonstrate the effectiveness of our method, we conduct comparative experiments on the DIML and RGB-NIR datasets, aiming to evaluate its performance against previous works. Tab. 3 presents the experimental results for two transfer tasks: RGB → Depth and RGB → NIR. Our method achieves remarkable accuracy improvements over SOCKET, with absolute gains of **+9.81%** and **+3.50%**, resulting in accuracies of **50.79%** and **90.48%**, respectively. These results clearly surpass those obtained by competing methods, establishing the superiority of our approach. Specifically, our proposed module, TGMB, exhibits superior performance compared to the source-only approach, demonstrating absolute improvements of **+17.75%** and **+4.72%** for the RGB → Depth and RGB → NIR tasks, respectively. These improvements highlight the effectiveness of TGMB in bridging the modality gap, which is facilitated

Table 4: Ablation study of loss components of TGMB on the **SUN RGB-D** dataset.

| $\mathcal{L}_{rec}$ | $\mathcal{L}_D^1$ | $\mathcal{L}_D^2$ | $\mathcal{L}_{IM}$ | K-v1 | K-v2 | Real | Xtion | Average |
|:---:|:---:|:---:|:---:|:---:|:---:|:---:|:---:|:---:|
| ✓ | | | | 30.45 | 9.57 | 6.28 | 12.56 | 14.72 |
| ✓ | ✓ | | | 31.72 | 10.81 | 6.84 | 13.91 | 15.82 |
| ✓ | | ✓ | | 31.96 | 10.65 | 6.12 | 13.20 | 15.48 |
| ✓ | ✓ | ✓ | | 33.07 | 11.13 | 7.00 | 14.71 | 16.48 |
| ✓ | ✓ | ✓ | ✓ | **34.50** | **12.26** | **8.74** | **15.82** | **17.76** |

Table 5: Ablation study of loss components of TGKT on the **SUN RGB-D** dataset.

| $\mathcal{L}_{kd}$ | $\mathcal{L}_f$ | $\mathcal{L}_{self}$ | K-v1 | K-v2 | Real | Xtion | Average |
|:---:|:---:|:---:|:---:|:---:|:---:|:---:|:---:|
| ✓ | | | 34.02 | 12.24 | 8.59 | 13.20 | 17.01 |
| ✓ | ✓ | | 37.84 | 17.17 | 15.98 | 25.66 | 24.16 |
| ✓ | | ✓ | 36.31 | 15.42 | 7.47 | 24.40 | 20.90 |
| ✓ | ✓ | ✓ | **41.49** | **18.76** | **15.66** | **28.62** | **26.13** |

by the inclusion of our proposed loss terms. Moreover, TGMB outperforms SHOT and SOCKET by margins of **+4.33%** and **+3.3%** in RGB → Depth tasks., respectively, further establishing its competitive advantages. Building upon the performance of TGMB, TGKT achieves additional improvements of **+6.49%** and **+8.89%** for the RGB → Depth and RGB → NIR tasks, respectively. This confirms the effectiveness of our proposed method in transferring knowledge from a source model with less reliable predictions to a target model, particularly when aided by paired TI data.

### 4.3 ABLATION STUDY AND ANALYSIS

In this section, we select the K-v1 dataset as the target modality and establish four transfer tasks to evaluate the performance of our proposed method. These tasks include K-v1 → K-v1, K-v2 → K-v1, Real → K-v1, and Xtion → K-v1 transfer tasks.

**Effectiveness of loss terms of TGMB.** To thoroughly examine the contributions of each loss component in our proposed TGMB module, we conduct ablation studies on the four transfer tasks. The results presented in Tab. 4 demonstrate satisfactory and consistent performance gains achieved by incorporating each loss term, highlighting their effectiveness in enhancing the overall performance. Compared to the source-only approach, the reconstruction loss $\mathcal{L}_{rec}$ yields a significant **+6.49%** performance gain. This improvement indicates the validity of estimating the source data distribution to mitigate the modality gap. Additionally, the discriminator losses $\mathcal{L}_D^1$ and $\mathcal{L}_D^2$ contribute to further improvements of **+1.10%** and **+0.76%**, respectively, by reducing the inter-modality and intra-modality gaps. The combination of these two losses results in a **+1.76%** performance gain, underscoring their effectiveness in generating source-like RGB images. Building upon these loss terms, the mutual information loss $\mathcal{L}_{IM}$ achieves an additional **+1.28%** performance gain. This improvement validates the efficacy of utilizing the source model to guide the generation of source-like RGB images, thereby facilitating bridging the modality gap.

**Effectiveness of loss components of TGKT.** To evaluate the efficacy of the loss components in our proposed TGKT and examine how their combination contributes to knowledge transfer, we conduct a comprehensive analysis. The results in Tab. 5 reveal the following key observations: (1) The feature matching loss $\mathcal{L}_f$ and the self-supervised pseudo-label loss $\mathcal{L}_{self}$ lead to notable improvements of **+7.15%** and **+3.89%** in accuracy, respectively. This highlights the effectiveness of knowledge transfer facilitated by the utilization of the paired TI data and the incorporation of a self-supervised pseudo-labeling approach. (2) By combining both losses, our proposed method achieves a significant improvement in knowledge transfer performance, reaching an accuracy of **26.13%**. This collective enhancement demonstrates the capacity of our method to effectively transfer knowledge from a source model with less reliable predictions to the target model, leveraging the paired TI data.

**Influence of $\alpha_d$ and $\alpha_{im}$.** To assess the impact of the weighting parameters $\alpha_d$ and $\alpha_{im}$ in the objective of TGMB, we conduct a comprehensive analysis. The results, presented in Tab. 6, illustrate the trade-off between the reconstruction loss $\mathcal{L}_{rec}$, discriminator loss $\mathcal{L}_D$, and mutual information

Table 6: Sensitivity of $\alpha_d$ and $\alpha_{im}$ evaluated on on the **DIML RGB-D** dataset.

| $\alpha_d$ | K-v1 | K-v2 | Real | Xtion | Average | $\alpha_{im}$ | K-v1 | K-v2 | Real | Xtion | Average |
|---|---|---|---|---|---|---|---|---|---|---|---|
| 0 | 30.45 | 9.57 | 6.28 | 12.56 | 14.72 | 0 | 33.07 | 11.13 | 7.00 | 14.71 | 16.48 |
| 0.1 | 30.62 | **12.24** | 6.52 | 12.72 | 15.53 | 0.1 | 33.39 | **12.40** | 8.51 | 14.79 | 17.27 |
| 0.5 | 32.27 | 11.29 | 6.92 | 13.67 | 16.04 | 0.2 | **34.50** | 12.26 | 8.74 | **15.82** | **17.76** |
| 1 | **33.07** | 11.13 | **7.00** | **14.71** | **16.48** | 0.5 | 30.60 | 10.89 | 6.68 | 14.56 | 15.68 |
| 5 | 32.21 | 10.92 | 6.43 | 11.62 | 15.30 | 1 | 27.98 | 10.33 | **8.82** | 14.94 | 15.82 |

Table 7: Sensitivity of $\beta_f$ and $\beta_{self}$ evaluated on on the **DIML RGB-D** dataset.

| $\beta_f$ | K-v1 | K-v2 | Real | Xtion | Average | $\beta_{self}$ | K-v1 | K-v2 | Real | Xtion | Average |
|---|---|---|---|---|---|---|---|---|---|---|---|
| 0.0 | 34.02 | 11.84 | 8.90 | 14.08 | 17.21 | 0.0 | 37.84 | 17.17 | 15.98 | 25.66 | 24.16 |
| 0.1 | 37.52 | 17.17 | 13.59 | 25.12 | 23.35 | 0.1 | 39.67 | 15.98 | 16.14 | 25.99 | 24.45 |
| 0.2 | **37.84** | **17.17** | 15.98 | 25.66 | **24.16** | 0.5 | 40.22 | 17.17 | **16.66** | 28.14 | 25.55 |
| 0.5 | 34.74 | 14.63 | 19.63 | **25.34** | 23.58 | 1.0 | **41.49** | 18.76 | 15.66 | **28.62** | **26.13** |
| 2.0 | 31.32 | 13.67 | **20.51** | 25.04 | 22.64 | 2.0 | 33.78 | **20.59** | 15.98 | 27.90 | 24.56 |

loss $\mathcal{L}_{IM}$, leading to several noteworthy findings: (1) After careful analysis, we select values of 1.0 and 2.0 for $\alpha_d$ and $\alpha_{im}$, respectively, as they demonstrate the optimal trade-off between the different loss components. (2) Note that when setting $\alpha_d$ to 5.0, the discriminator loss $\mathcal{L}_D$ may adversely impact the performance on the Xtion $\rightarrow$ K-v1 transfer task. This can be attributed to the fact that TGMB primarily focuses on reducing the modality gap, potentially disregarding the importance of reconstruction performance. (3) The lack of ground truth of target data and the modality gap poses challenges in generating reliable predictions from the source model for the source-like images. Consequently, increasing the weight of the mutual information loss $\mathcal{L}_{IM}$ may inadvertently misguide the translation process, resulting in the generation of source-like RGB images that do not align with the source data distributions.

**Influence of $\beta_f$ and $\beta_{self}$.** To assess the effectiveness of knowledge transfer, we conduct a hyperparameter analysis on $\beta_f$ and $\beta_{self}$, as presented in Tab. 7. In addition to validating the effectiveness of our method and determining the optimal trade-off parameters $\beta_f$ and $\beta_{self}$, the results yield several important observations: (1) Increasing the value of $\beta_f$ may lead to degraded performance. This can be attributed to the fact that feature matching between the paired TI data facilitates knowledge transfer from the source to target modalities. However, it may also have a negative impact on the training for the task of interest when the intra-modality gap between the TR and TI data is large. Consequently, it becomes crucial to select a trade-off parameter that strikes a balance between reducing the modality gap and training the model for the task of interest. (2) The self-supervised pseudo-labeling approach may adversely affect performance, particularly in scenarios where the predictions of the TR target data are less reliable. *More ablation study results and discussion can be found in suppl. material.*

## 5 CONCLUSION AND FUTURE WORK

In this work, we presented a novel yet concise framework to tackle a challenging source-free cross-modal knowledge transfer problem. Our approach capitalizes on the unexplored potential of task-irrelevant data to enhance the knowledge transfer for the task of interest. Specifically, our proposed method leverages task-irrelevant data to facilitate translating the TR target data into the TR source-like RGB images and transferring knowledge from the source model with less reliable predictions to the target model. Extensive experiments conducted on three datasets verify that our method achieves the state-of-the-art performance compared to previous methods.

**Limitation and future work.** One limitation of our method lies in its non-end-to-end framework, which signifies that there are certain aspects requiring further development. To address this limitation, our future endeavors will focus on updating the modality bridging and knowledge transfer processes in an iterative manner. This iterative approach will enable us to devise more effective solutions for the problem at hand. Furthermore, our future research will delve into investigating the utilization of the task-irrelevant data as an auxiliary tool to facilitate knowledge distillation for the task of interest.

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

# A APPENDIX

## ABSTRACT

Due to space limitations in the main paper, we provide more details of the proposed method and experimental results in the supplementary material. In Sec.B, we delve into a discussion regarding our proposed method. Sec.C provides comprehensive information about the dataset used in our experiments, including details about the class split. In Sec.D, we present in-depth descriptions of the individual modules and the training procedures employed in our work. Sec.E introduces self-supervised pseudo-labeling approach as introduced by Liang et al. (2020b). To facilitate understanding and reproducibility, we provide a pseudo algorithm of our proposed method in Sec.F. Sec.G presents an overview of potential future research directions. Finally, Sec.H shows the boarder impact of proposed method. *Our project code will be publicly available upon acceptance.*

# B DISCUSSION

## B.1 COMPARISON WITH IMAGE TRANSLATION METHOD

To demonstrate that *using existing image translation methods Razavi et al. (2019); Esser et al. (2021); Van Den Oord et al. (2017) alone does not meet our specific needs as they primarily aim to reconstruct natural-looking images rather than optimal source-like RGB images for the source model,* we conduct a comparative analysis. To this end, we employ VQ-GAN Esser et al. (2021) pretrained with TI source images to reconstruct the TR target images. Additionally, we utilize our proposed TGMB to generate TR source-like images. For clarity, we illustrate this comparison using the K-v1 → K-v1 task as an example. Firstly, we train VQ-GAN with the TI source data to enable accurate reconstruction of the TI source images. Subsequently, we input the TR source data into VQ-GAN, generating the reconstructed TR images. These reconstructed TR images are presented in in Fig.3.

To ensure a fair comparison, we also utilize our proposed TGMB approach to generate TR source-like images. These source-like images are depicted in Fig. 4. We then input both the reconstructed TR images obtained via VQ-GAN and the source-like images generated by TGMB into the pre-trained source model. Upon evaluation, the model utilizing the reconstructed TR images via VQ-GAN achieves an accuracy of **29.21%**. In contrast, when employing the source-like images generated by TGMB, the accuracy significantly improves to **34.50%**. This comparison effectively demonstrates the efficacy of our proposed TGMB approach in generating source-like images that bridge the modality gap. Note that while the generated source-like images may not be visually recognizable by humans, they possess similar representations and outputs to the source domain data when processed by convolutional neural networks. In contrast, although the reconstructed images obtained via VQ-GAN appear natural, they are sub-optimal for addressing downstream tasks, emphasizing the importance of our proposed TGMB in generating suitable source-like images.

## B.2 DETAILS OF COMPUTATIONAL COST

The generation module entails a computational cost of 190.841M/MACs, accompanied by a parameter count of 2.851k. In the training progress of TGMB, since the translation layer consists of a few layers and discriminators consist of three fully connected layers, the computational cost of the generation module could be negligible compared with training the target model. Note that in the inference, we only utilize the target model to label the target data without other modules like the generation module. Therefore, testing the model is much efficient without computational cost of generation module.

## B.3 NOVELTY OF THIS WORK

Our novelty is to "unlock the potential of the paired TI data to accurately estimate the source data distribution and facilitate knowledge transfer from the source model to the target model for the unlabeled TR target data". Our work introduces a novel problem that has been overlooked by previous studies. We have meticulously devised a distinct framework that harnesses TI data to facilitate the

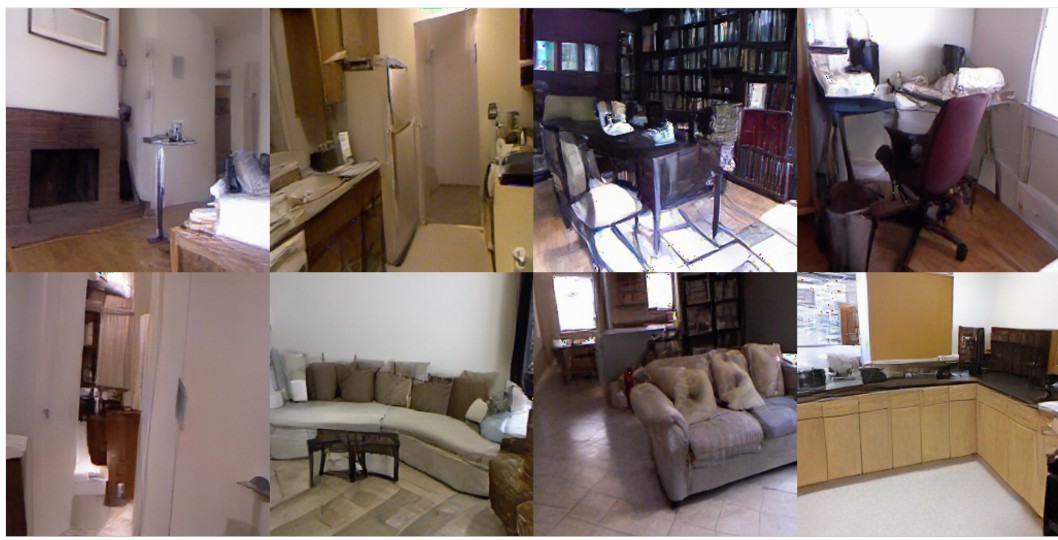

Figure 3: Reconstructed TR images via VQ-GAN on the K-v1 → K-v1 transfer task.

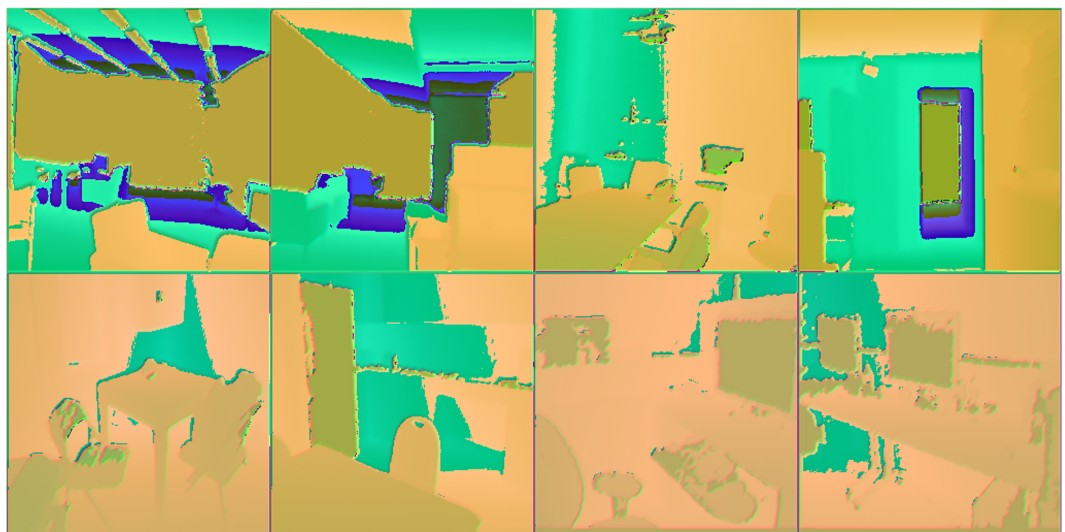

Figure 4: Generated TR source-like images via TGMB on the K-v1 → K-v1 transfer task.

estimation of TR source-like data and transfer of cross-modal knowledge. To achieve these aims, we propose a specific module TGMB to generate TR source-like images via utilizing paired TI data to guide this translation and further utilize a pre-trained source model to control the generation. Addressing specific research challenges is inherent to our approach. Notably, we propose targeted solutions to reduce both intra-modality and inter-modality gaps. This strategic approach leads to the generation of source-like RGB images that effectively bridge the modality gap. Subsequently, upon obtaining these TR source-like RGB images, we advocate for an approach that optimally harnesses TI data to facilitate knowledge transfer from a source model with potentially less reliable predictions to a target model that addresses the task of interest. Note that our method departs from conventional knowledge distillation methods by emphasizing the minimization of distances between paired TI data within the feature space, rather than relying on straightforward knowledge transfer between TR data. Additionally, we incorporate self-supervised pseudo-labeling to tackle specific challenges where predictions of TR source-like data is less reliable. Our proposed framework's efficacy is reinforced by compelling results and visualizations. These outcomes empirically demonstrate the effectiveness of our approach and its potential to address the identified challenges.

### B.4 DETAILS OF TRAINING DISCRIMINATORS

Each discriminator receives two feature vectors. The inputs of discriminator $D_1$ are the features of TI source data and translated TI source-like data which is generated through the translation layer. The inputs of the discriminator $D_2$ comprise the feature vectors from both translated TI and TR source-like data.

### B.5 ROLE OF TI DATA

Our research pertains to the complex task of estimating TR source data, a challenge that underpins our work. To estimate the TR source data, we propose the TGMB module to generate TR source-like data with paired TI data guidance. Our rationale and strategies for modeling the TR source data are expounded below. Since we have paired TI data, we propose a translation layer to translate the TI target data into TI source-like data with reconstruction loss and inter-modality minimization loss. Since TI and TR source data is from different distributions, we aspire for the translation layer to not only function optimally with TI data but to seamlessly accommodate TR data as well. To this end, we propose to minimize the intra-modality gap between generated TI and TR source data. Finally, the translation layer can adeptly generate both TI and TR source data. To ensure a robust alignment between the generated TR source-like images and the original source distribution, we capitalize on pre-existing pre-trained source models and mutual information loss to guide the translation.

### B.6 DETAILS OF TR/TI SPLIT

The context of our entire work is rooted in the setting, where the availability of paired TI data serves as the backdrop. This paired TI data plays a crucial role in facilitating the transfer of knowledge across different modalities – from a source modality to a distinct target modality – in situations where access to TR source data is unavailable. The rationale behind utilizing paired TI data, as elucidated in the first work SOCKET, is as follows: "SOCKET addresses a challenge within the context of cross-modal knowledge transfer. It operates under the premise that only (a) source models trained for the task of interest (TOI), and (b) unlabeled data within the target modality are accessible, necessitating the construction of a model for the same TOI. A key aspect of this problem lies in the assumption that no data from the source modality for TOI is accessible. This setup holds significance in scenarios where memory and privacy constraints preclude the sharing of training data from the source modality, permitting only the exchange of trained models. In response, SOCKET is proposed to bridge the disparity between the source and target modalities. In this work, It is demonstrated that leveraging an external dataset of source-target modality pairs, unrelated to TOI – designated as TI data – can facilitate the learning of an effective target model by reducing the feature discrepancy between the two modalities". Therefore, this first work SOCKET provides the foundational context for our entire work. Building upon the limitations and issues identified in previous research, we propose a research problem and subsequently propose a novel and efficacious framework to address this problem, thereby achieving state-of-the-art performance.

### B.7 COMPARISON WITH SOCKET

In this work, we propose a research question based on SOCKET, and observe that the paired TI data plays a crucial role in bridging the modality gap. The key factor of improving performance is utilizing paired TI data as a guidance to estimate the source data distribution and cross-modal knowledge transfer. Specifically, within the context of estimating TR source data, we utilize paired TI data to diminish the inter-modality and intra-modality gaps to facilitate the generation of TR source-like data. Moreover, we fully exploit the knowledge of the pre-trained source model to ensure that the translated TR source-like images closely resemble the source data distribution required for effective knowledge transfer. In the process of cross-modality knowledge transfer, we utilize paired TI data to align the feature representations of the two modalities. Moreover, we further adopt a self-supervised pseudo-labeling method to solve the problem that predictions made by the source model on the TR source-like data are less accurate.

## B.8 GENERATION OF PAIRED TR DATA

In this work, the absence of direct access to TR source data is acknowledged. This renders direct knowledge transfer from the source model to the target modality data unfeasible. Therefore, we propose a TGMB module to translate the TR target data into TR source-like data with TI data-guidance. It is imperative to recognize that the TR target data and its corresponding TR source-like data are in fact meticulously paired. Since TR target data is unlabeled, we propose TGKT to transfer knowledge from the source model to the target modality data based on generating TR source-like data. Then we utilize KL-divergence to facilitate the knowledge transfer between different modalities. Addressing the challenges posed by modality gaps and the reduced predictive reliability of TR source-like data from the source model, we have devised $\mathcal{L}_f$ and $\mathcal{L}_{self}$. These mechanisms serve to navigate these complexities within the framework of paired TI data-guidance.

## B.9 SELECTION OF TRANSLATION LAYER

Since the utilization of a diffusion model for image generation could potentially enhance computational efficiency, this approach might not align seamlessly with our specific requirements. Diffusion models primarily aim to reconstruct natural-looking images, a goal that diverges from our objective of producing optimal source-like RGB images for the source model's inputs. Training a diffusion model necessitates a substantial volume of paired images, which contrasts with the DIML and RGB-NIR datasets that comprise merely a few hundred images. This disparity led us to forego the utilization of conventional image generation methods for generating TR source-like images. Central to our endeavor is the domain of source-free domain adaptation. In this context, our employment of the generation method is geared towards producing TR source-like data. Our primary focus is on enhancing effective generation by decreasing intra-modality and inter-modality gaps. Additionally, the integration of an information maximization loss bolsters the generation process. Within this work, we present a translation layer distinguished by its simplicity yet remarkable efficacy. The translation layer, comprised of just a few convolutional layers, plays a pivotal role in bridging the modality gap.

## C DATASET

To testify the versatility of the proposed method, we conduct experiments on several public visual datasets: SUN RGB-D Song et al. (2015), DIML RGB+D Cho et al. (2021), and RGB-NIR Brown & Süsstrunk (2011).

**SUN RGB-D** is an indoor scene benchmark containing 10335 RGB-D image pairs which are captured by four sensors, including Kinect version1 (K-v1), Kinect version2 (K-v2), Intel RealSense (Real), and Asus Xtion (Xtion). We regard the images taken by different sensors from different domains. The whole dataset consists of 45 classes, in which we take 17 common classes as TR classes and the left 28 classes as TI classes. To obtain the source models from different domains, we train them via RGB images from TR classes while the target modality data are the corresponding depth images from TR classes. We build sixteen tasks to evaluate our method.

**DIML RGB-D** is comprised of over 200 indoor/outdoor scenes while we only use 2112 RGB-D image pairs with 18 indoor scenes for evaluation. We split the 18 classes into two components: 6 classes as TR data and the remaining 12 classes as TI data. In each image pair, the RGB and depth are regarded as the source and target, respectively. We report the performance of one transfer task: RGB → Depth.

**RGB-NIR** contains 477 RGB and Near-Infrared (NIR) image pairs of 9 scene categories. The images were captured using separate exposures from modified SLR cameras, using visible and NIR filters. The whole dataset is split into TR data with 6 classes and TI data with 3 classes, and the knowledge is transferred via a single source from RGB and NIR.

## C.1 SUN RGB-D

The SUN RGB-D dataset with 45 classes is split into TR data and TI data. The TR data contains 17 common scenes which share among the four domains, including *bathroom, classroom, computer room, conference room, corridor, discussion area, home office, idk, kitchen, lab, living room, office,*

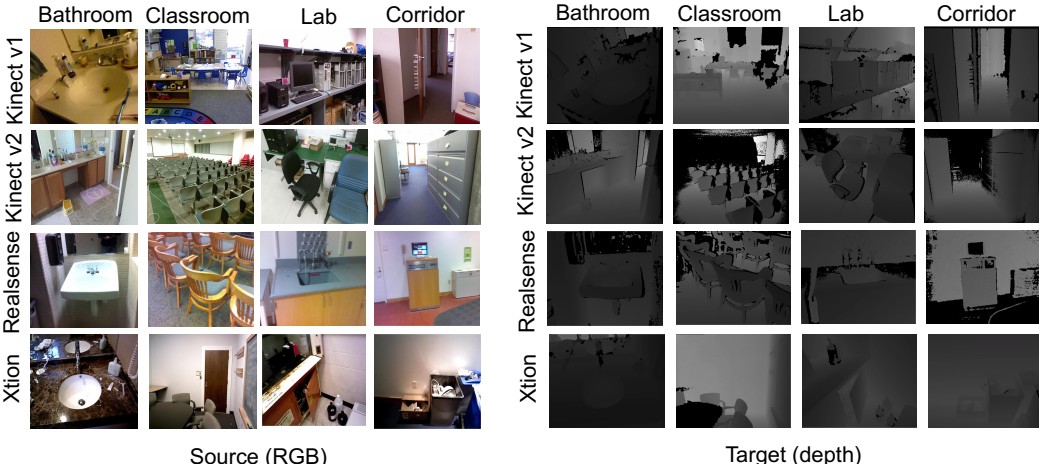

Figure 5: **SUN RGB-D TR sample images**. The TR data is from the 17 scenes out of 45 classes across two modalities and 4 out of 17 classes are shown here. The source RGB images are discarded after finishing training the source model, and the labels are unavailable for the target depth samples in our settings.

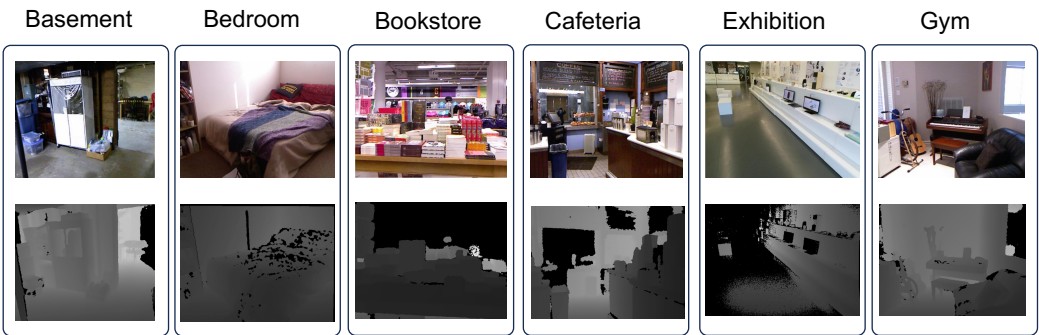

Figure 6: **SUN RGB-D TI sample images**. We show some samples of TI images which comprise 28 classes, and list 6 classes, each with paired RGB and depth samples here.

*office kitchen, printer room, reception room, rest space, study space*. The TR samples in SUN RGB-D dataset are shown as Fig.5.

The data with the remaining 28 classes is categorized into TI data, where the remaining classes are *basement, bedroom, book store, cafeteria, coffee room, dancing room, dinette, dining area, dining room, exhibition, furniture store, gym, home, study, hotel room, indoor balcony, laundromat, lecture theatre, library, lobby, mail room, music room, office dining, play room, reception, recreation room, stairs, storage room*. The TI samples in SUN RGB-D dataset are shown as Fig.6.

## C.2    DIML RGB-D

The 6 classes used as TR data in DIML RGB-D dataset are *bathroom, classroom, computer room, kitchen, corridor, living room*. The TR samples in DIML RGB-D dataset are shown as Fig.7.

The left 12 scenes regarded as the TI data are *bedroom, billiard hall, book store, cafe, church, hospital, laboratory, library, meeting room, restaurant, store, warehouse*. The TI samples in DIML RGB-D dataset are shown as Fig.8.

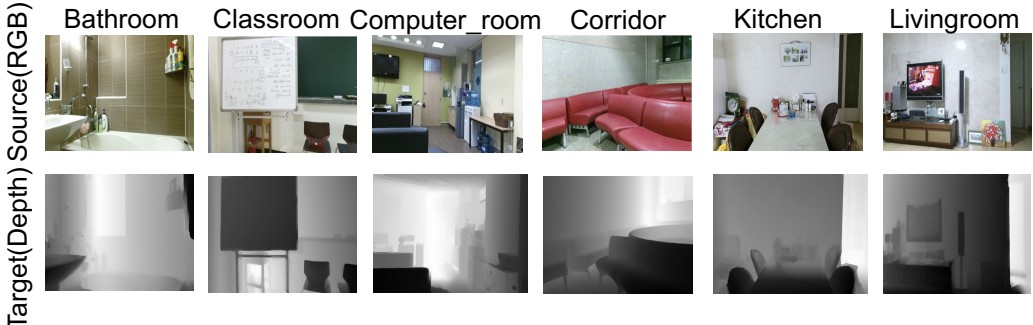

Figure 7: **DIML TR sample images**. We show all the scenes in TR data, which comprises 6 classes. The source RGB samples are not available after training, and the target labels are not accessible during the whole process.

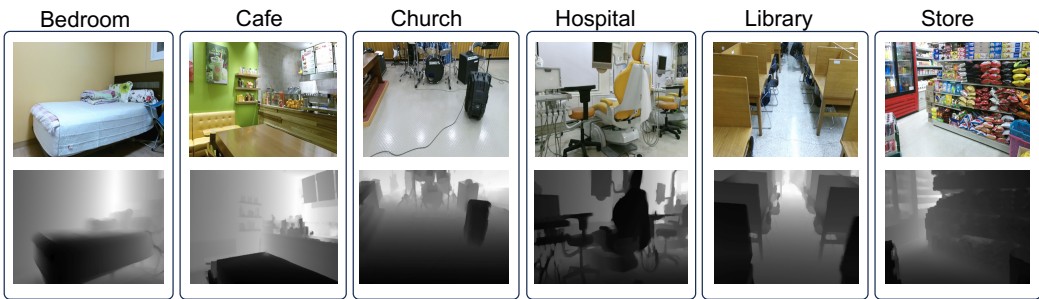

Figure 8: **DIML TI sample images**. We show 6 out of 12 classes for TI data, each with paired RGB and depth samples here.

## C.3   RGB-NIR

The TR data in RGB-NIR dataset contains 6 classes, which are *country, field, indoor, mountain, street, water*. The TR samples in RGB-NIR dataset are shown as Fig.9

The remaining 3 classes used as TI data are *forest, old building, urban*. The TI samples in RGB-NIR dataset are shown as Fig.10

## D   IMPLEMENTATIONS

In our experimental setup, ResNet-50 He et al. (2016) serves as the feature extractor module for training both the source and target models. The architectural modifications we employ are in line with the approach proposed in Liang et al. (2020a); Xu et al. (2019). Specifically, we replace the last fully connected layer with a bottleneck layer and a task-specific classifier layer. Additionally, we incorporate batch normalization layer Ioffe & Szegedy (2015) after the bottleneck layer and apply weight normalization in the final layer. To reduce the inter-modality and intra-modality gaps within the models, we introduce two discriminators that are designed based on adversarial learning Goodfellow et al. (2020). Each discriminator consists of three fully connected layers, with a Rectified Linear Unit (ReLU) Nair & Hinton (2010) activation function applied after the first two layers. The translation module, crucial for our framework, is constructed by combining a fully connected layer, a batch normalization layer, and a convolutional layer, adopting a UNET Ronneberger et al. (2015) architecture. Specifically, we utilize the first DoubleConv module and OutConv module to establish the translation module, which facilitates the conversion process between different modalities. To generate images that closely resemble the inputs of the source model, we impose constraints for the outputs of the translation module using the sigmoid Dubey et al. (2022) and normalize functions. These constraints ensure the production of source-like images during the generation process, maintaining consistency with the desired outputs.

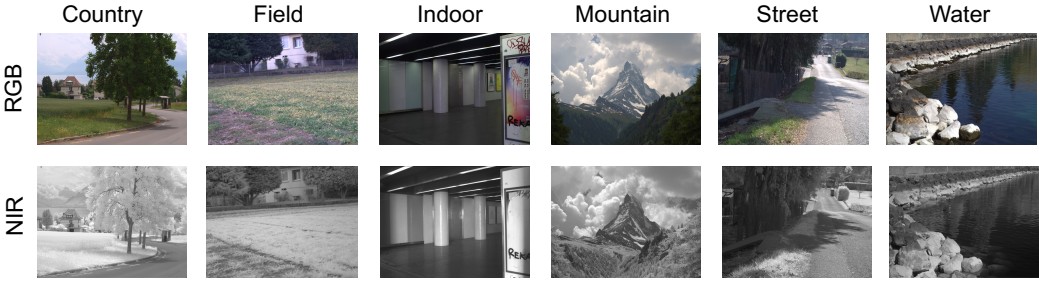

Figure 9: **RGB-NIR TR sample images**. There are 6 classes for TR data, and we show all the scenes here.

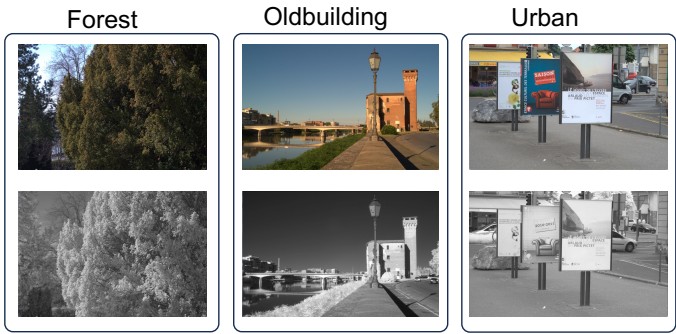

Figure 10: **RGB-NIR TI sample images**. We show all the 3 classes for TI data, and they all RGB and depth paired image samples.

Furthermore, each framework is trained using the back-propagation algorithm, a widely adopted technique in deep learning. Specifically, for training the source model, we employ stochastic gradient descent with a momentum value of 0.9 and weight decay set to $10^{-3}$. The learning rate is set at $10^{-2}$ for the bottleneck and classifier layers, while the backbone is trained at a rate of $10^{-3}$. Additionally, we incorporate a learning rate scheduling strategy, as described in Ganin & Lempitsky (2015), where the initial rate is exponentially decayed as the learning progresses. This strategy aids in optimizing the training process. During the training process of the translation layer in the TGMB, we utilize stochastic gradient descent with a momentum value of 0.9, weight decay set to $10^{-3}$, and a learning rate of $10^{-3}$. For training the discriminators, we employ the Adam optimizer, a popular optimization algorithm, and set the learning rate to $10^{-4}$. It is important to note that in the TGMB process, the parameters of the source model are fixed. In the TGKT process, we leverage both the translation module and source model to teach the target model. The architecture of the target model is identical to that of the source model. For training the target model, we also utilize stochastic gradient descent with a momentum value of 0.9 and weight decay set to $10^{-3}$. The learning rate is set at $10^{-2}$ for the bottleneck and classifier layers, while the backbone is trained at a rate of $10^{-3}$. Throughout the training of the source model, TGMB, and TGKT, we employ a batch size of 32, a common practice in deep learning experiments. To facilitate the implementation of our experiments, we utilize the PyTorch framework Paszke et al. (2019), a widely-used deep learning library. For a comprehensive understanding of the algorithm, we provide the detailed algorithm in Sec. F.

## E  SELF-SUPERVISED PSEUDO-LABELING

To obtain more accurate pseudo-labels, we follow the method as the work Liang et al. (2020b)Ahmed et al. (2021) and adopt a self-supervised clustering strategy to refine the wrong predictions. Firstly, we calculate the cluster centroid of each class by target model for the whole target data at iteration 0 as follows:

$$c_k^{(0)} = \frac{\sum_{i=1}^{n_t} \delta_k\left(\hat{G}_t\left(x_i^t\right)\right) \hat{F}_t\left(x_i^t\right)}{\sum_{i=1}^{n_t} \delta_k\left(\hat{G}_t\left(x_i^t\right)\right)}, \tag{11}$$

where $\hat{G}_t = C(\hat{F}_t(x_i^t))$ denotes the target model from the last iteration, $\hat{F}_t$ represents the feature extractor from the last iteration and $C$ is the classifier which is fixed during the whole process. The calculated centroids are capable of representing the distribution of different categories of target domains more robustly and reliably. Next, we assign the class label $k$ as the pseudo-label for the $i$-th target feature $\hat{F}_t(x_t^i)$ when its nearest neighbor is the $k$-th centroid. The pseudo-label at iteration 0 is calculated as:

$$\hat{y}_t^{(0)} = \arg\min_k \left\| \hat{F}_t(x_t^i) - c_k^{(0)} \right\|_2^2, \tag{12}$$

After that, we continue to compute the target centroids based on the updated pseudo-labels by repeating the steps:

$$c_k^{(1)} = \frac{\sum_{i=1}^{n_t} \mathbf{1}\left\{\hat{y}_t^{(0)} = k\right\} \hat{F}_t\left(x_i^t\right)}{\sum_{i=1}^{n_t} \mathbf{1}\left\{\hat{y}_t^{(0)} = k\right\}}, \tag{13}$$

$$\hat{y}_t^{(1)} = \arg\min_k \left\| \hat{F}_t(x_t^i) - c_k^{(1)} \right\|_2^2, \tag{14}$$

where $\mathbf{1}\{.\}$ denotes an indicator function that equals 1 when the inside argument is true. Here $\hat{y}_t$ represents the self-supervised pseudo-labels as they are produced by the centroids calculated in an unsupervised manner. We reiterate updating the pseudo-labels regularly based on Eq.(3) and (4) until reaching certain iterations. Finally, the pseudo-label cross-entropy loss $\mathcal{L}_{self}$ is calculated as follow:

$$\mathcal{L}_{\text{self}} = \frac{1}{n_t} \sum_{i=1}^{n_t} \mathbf{1}_{[k=\hat{y}_t]} \log\left(\delta_k\left(\mathbf{C}\left(\mathbf{F}_t(x_i^t)\right)\right)\right). \tag{15}$$

## F   ALGORITHM

The pseudo algorithms of the proposed TGMB and TGKT are shown in Algorithm. 1 and Algorithm. 2, respectively.

## G   FUTURE WORK

The utilization of TI data for facilitating knowledge transfer presents challenges due to its out-of-distribution nature when compared to the TR (Target Reconstruction) data. However, in our work, we demonstrate that TI data can indeed be leveraged to minimize the gap between the source and target models. It is important to note that the impact of TI data on training can vary depending on the magnitude of the intra-modality gap between the TR and TI data and the emphasis placed on reducing this gap versus improving supervised performance.

Additionally, the effectiveness of the self-supervised pseudo-labeling approach relies on the reliability of the predictions generated by the target model for the TR target data. In cases where the predictions for TR data from the source model are less reliable, the self-supervised pseudo-labeling approach can enhance the performance of the target model. However, it is crucial to acknowledge that this approach may also have adverse effects, particularly when the predictions of TR target data from the target model are unreliable. This implies that the pseudo-labeling process generates less reliable predictions for TR data, consequently misleading the training of the target model.

Considering the challenges associated with utilizing TI data for knowledge transfer and the potential limitations of the self-supervised pseudo-labeling approach, future research efforts will be directed towards exploring the utilization of task-irrelevant data to transfer knowledge between cross-modal TR data. Additionally, we intend to investigate approaches that ensure the reliable guidance of the target model's training process using the self-supervised pseudo-labeling approach, while effectively

---

**Algorithm 1** Task-irrelevant data-Guided Modality Bridging

---

1: **Input**: $x^s_{TI_i}$, $x^t_{TI_i}$, $x^t_i$; max iterations: $\mathbf{I}$; model: $\mathbf{F}_s$, $\mathbf{C}$, $\mathbf{T}$, $\mathbf{D}_1$, $\mathbf{D}_2$.
2: **for** i to $\mathbf{I}$ **do**
3:     Utilize translation module $\mathbf{T}$ to generate source-like TR and TI images: $\mathbf{T}(x^t_i)$, $\mathbf{T}(x^t_{TI_i})$.
4:     Calculate the reconstruction loss using paired TI images:
    $\mathcal{L}_{rec} = \frac{1}{n_{TI}} \sum_{i=1}^{n_{TI}} \left\| x^s_{TI_i} - \mathbf{T}(x^t_{TI_i}) \right\|^2$.
5:     Decrease the inter-modality and intra-modality gaps with two discriminators:
    $\mathcal{L}^1_D = \frac{1}{n_{TI}} \sum_{i=1}^{n_{TI}} \log(\mathbf{D}_1(\mathbf{F}_s(x^s_{TI_i})) + \frac{1}{n_{TI}} \sum_{i=1}^{n_{TI}} \log(1 - \mathbf{D}_1(\mathbf{F}_s(\mathbf{T}(x^t_{TI_i}))))$,
    $\mathcal{L}^2_D = \frac{1}{n_t} \sum_{i=1}^{n_t} \log(\mathbf{D}_2(\mathbf{F}_s(\mathbf{T}(x^t_i))) + \frac{1}{n_{TI}} \sum_{i=1}^{n_{TI}} \log(1 - \mathbf{D}_2(\mathbf{F}_s(\mathbf{T}(x^t_{TI_i}))))$,
    $\mathcal{L}_D = \mathcal{L}^1_D + \mathcal{L}^2_D$.
6:     To improve the suitability of the TR source-like images for the source model, we maximize the mutual information of TR source-like images:
    $\mathcal{L}_{ent} = -\frac{1}{n_t} \left[ \sum_{i=1}^{n_t} \delta_k(\mathbf{C}(\mathbf{F}_s(\mathbf{T}(x^t_i)))) \log \delta_k(\mathbf{C}(\mathbf{F}_s(\mathbf{T}(x^t_i)))) \right]$,
    $\mathcal{L}_{div} = -\sum_{k=1}^{K} \bar{p}_k \log \bar{p}_k$,
    $\mathcal{L}_{IM} = \mathcal{L}_{ent} - \mathcal{L}_{div}$.
7:     Compute the total objective for TGMB:
    $\mathcal{L}_{TGMB} = \mathcal{L}_{rec} + \alpha_d \mathcal{L}_D + \alpha_{im} \mathcal{L}_{IM}$.
8:     Update the translation module $\mathbf{T}$ with $\mathcal{L}_{TGMB}$.
9:     To update the two discriminators, we calculate the discriminator loss as:
    $\mathcal{L}^1_D = \frac{1}{n_{TI}} \sum_{i=1}^{n_{TI}} \log(\mathbf{D}_1(\mathbf{F}_s(x^t_{TI_i})) + \frac{1}{n_{TI}} \sum_{i=1}^{n_{TI}} \log(1 - \mathbf{D}_1(\mathbf{F}_s(\mathbf{T}(x^s_{TI_i}))))$,
    $\mathcal{L}^2_D = \frac{1}{n_{TI}} \sum_{i=1}^{n_{TI}} \log(\mathbf{D}_2(\mathbf{F}_s(\mathbf{T}(x^t_{TI_i})))) + \frac{1}{n_t} \sum_{i=1}^{n_t} \log(1 - \mathbf{D}_2(\mathbf{F}_s(\mathbf{T}(x^t_i))))$,
10:    Update the discriminators $\mathbf{D}_1$ and $\mathbf{D}_2$ with $\mathcal{L}^1_D$ and $\mathcal{L}^2_D$, respectively.
11: **end for**
12: **return** translation module $\mathbf{T}$.
13: **End**.

---

**Algorithm 2** Task-irrelevant data-Guided Knowledge Transfer

---

1: Input: $x^s_{TI_i}$, $x^t_{TI_i}$, $x^t_i$; max iterations: $\mathbf{I}$; model: $\mathbf{F}_s$, $\mathbf{F}_t$, $\mathbf{C}$, $\mathbf{T}$.
2: **for** i to $\mathbf{I}$ **do**
3:     Transfer knowledge from the source model to the target model using paired TR target data and source-like images:
    $\mathcal{L}_{kd} = \frac{1}{n_t} \sum_{i=1}^{n_t} KL \left( \delta_k(\mathbf{C}(\mathbf{F}_t(x^t_i)) \| \delta_k(\mathbf{C}(\mathbf{F}_s(\mathbf{T}(x^t_i)))) \right)$.
4:     Utilize pried TI data to facilitate knowledge transfer between the source and target model:
    $\mathcal{L}_f = \frac{1}{n_{TI}} \sum_{i=1}^{n_{TI}} \left\| \mathbf{F}_s(x^s_{TI_i}) - \mathbf{F}_t(x^t_{TI_i}) \right\|^2$.
5:     Apply self-supervised pseudo-labeling approach to enable the target model to learn from its own predictions:
    $\mathcal{L}_{self} = \frac{1}{n_t} \sum_{i=1}^{n_t} \mathbf{1}_{[k=\hat{y}_t]} \log \delta_k(\mathbf{C}(\mathbf{F}_t(x^t_i)))$.
6:     The total objective of the TGKT process is defined as:
    $\mathcal{L}_{TGKT} = \mathcal{L}_{kd} + \beta_f \mathcal{L}_f + \beta_{self} \mathcal{L}_{self}$.
7:     Update the target model $\mathbf{F}_t$ with the loss $\mathcal{L}_{TGKT}$.
8: **end for**
9: **return** target model $\mathbf{F}_t$.
10: **End**.

---

transferring knowledge from the source model (teacher) to the target model (student). These endeavors aim to overcome the aforementioned challenges and advance the field of knowledge transfer in cross-modal settings.

## H   BOARDER IMPACT

This paper represents a pioneering contribution in the field of source-free cross-modal knowledge transfer by incorporating task-irrelevant data to facilitate the transfer of knowledge for task-relevant data. This novel approach fills a critical gap in the current research landscape, as it addresses the

challenge of knowledge transfer between different modalities without requiring task-relevant data. By leveraging task-irrelevant data as a valuable resource, our proposed method has the potential to impact the development of cross-modality large-scale models for the task of interest in real-world applications.

