# OpenReview forum: "Source-free Cross-modal Knowledge Transfer by Unleashing the Potential of Task-Irrelevant Data"
_ICLR.cc/2024/Conference — ICLR 2024 Conference Withdrawn Submission_

### Official Review · Reviewer_5PgS · 2023-10-31

**Soundness:** 2 fair
**Presentation:** 3 good
**Contribution:** 2 fair
**Rating:** 5
**Confidence:** 4

**Summary:**

The paper addresses the challenge of source-free cross-modal knowledge transfer, specifically transferring knowledge between modalities such as RGB and depth or infrared without task-relevant (TR) source data. While previous methods directly matched features from task-irrelevant (TI) data to bridge the modality gap, this work proposes using the paired TI data to more accurately estimate the source data distribution. The authors introduce two modules: Task-irrelevant data-Guided Modality Bridging (TGMB) and Task-irrelevant data-Guided Knowledge Transfer (TGKT). TGMB translates target modality data to resemble source modality data, addressing both inter-modality and intra-modality gaps. Meanwhile, TGKT transfers knowledge between source and target models using a self-supervised pseudo-labeling technique due to the lack of labels for TR target data.

**Strengths:**

1. The study showcases superior performance relative to the current state-of-the-art, as presented by Ahmed et al.

2. The manuscript is articulated with clarity, ensuring it is both accessible and easy to comprehend.

3. Experimental findings are both compelling and indicative of the method's potential.

4. Building upon the foundation laid by Ahmed et al., this research harnesses task-irrelevant data. It achieves this by estimating the source data distribution, which in turn helps to bridge the modality gap. This facilitates more effective knowledge transfer from the source model to the target model, especially concerning unlabeled Task Relevant target data.

**Weaknesses:**

1. The most pressing reservation regarding this paper pertains to its novelty. The presented research appears to be an incremental advancement, largely built on the foundations set by Ahmed et al. in the domain of source-free cross-modal knowledge transfer. Its central contribution seems limited to the introduction of a translation module, potentially underscoring a lack of significant technical advancement.


2. While the authors duly note the implications of the translation network—namely, increased parameters and extended training durations—the evaluation predominantly hinges on accuracy metrics. Given the similarity of the task to that of Ahmed et al., a holistic assessment encompassing accuracy, memory, and time overhead during training would provide a more equitable comparison between the baseline and the current approach.

**Questions:**

1. Within the complete learning process, how do the authors plan to assess the appropriateness of the reconstructed (or translated) image in relation to cross-modal transfer?

2. Is it feasible to integrate the translation network into the standard SOCKET framework? Should this be possible, incorporating the translation framework within SOCKET ought to be considered as a benchmark for comparison.

---

### Official Review · Reviewer_vgxe · 2023-11-01

**Soundness:** 2 fair
**Presentation:** 3 good
**Contribution:** 1 poor
**Rating:** 3
**Confidence:** 3

**Summary:**

This paper proposes a framework to unlock the potential of paired TI data for enhancing source-free cross-modal knowledge transfer. They design two components: a Task-irrelevant data-Guided Modality Bridging (TGMB) module and a Task-irrelevant data-Guided Knowledge Transfer (TGKT) module.

**Strengths:**

This paper claim that the paired TI data could be utilized to effectively estimate the source data distribution and better facilitate knowledge transfer to the target modality. TI data may be important for this task.
The paper is easy to follow
This method achieves the state-of-the-art performance on three datasets (RGB-to-depth and RGB-to-infrared).

**Weaknesses:**

Firstly, I have reservations about the proposed task as it appears to be a forced combination of depth maps, infrared images, RGB, and source-free domain adaptation. I am uncertain about the motivation behind considering such a scenario, as it seems like an arbitrary merging of two distinct tasks.

Regarding the technical approach, I find some aspects to be incremental. Specifically, the proposed translation scheme has been extensively used in the context of source-free domain adaptation [1][2][3]. Additionally, the use of additional TI (unrelated) data for distillation in cross-modal distillation and similar approaches in semi-supervised learning have been widely explored. It seems that the paper does not yield particularly novel or interesting conclusions in the context of this task. Furthermore, there is a lack of discussion on how the cross-modal representations are better suited for source-free domain adaptation.

The paper lacks comparisons with recent source-free domain adaptation papers [1][2][3][4][5][6], which is essential for evaluating the proposed method's performance against state-of-the-art approaches in this field.

There are some writing issues and details that need attention. For example, the arrows in Table 3 are not aligned properly, which affects the overall readability and clarity.

[1] Generative Alignment of Posterior Probabilities for Source-free Domain Adaptation, WACV2023

[2] Source-free Domain Adaptation via Avatar Prototype Generation and Adaptation, IJCAI2021

[3] Source-Free Domain Adaptation via Distribution Estimation, CVPR2022

[4] Guiding Pseudo-labels with Uncertainty Estimation for Source-free Unsupervised Domain Adaptation, CVPR2023

[5] Class Relationship Embedded Learning for Source-Free Unsupervised Domain Adaptation, CVPR2023

[6] Attracting and dispersing: A simple approach for source-free domain adaptation, NeurIPS2022

**Questions:**

1. The proposed task appears to be a forced combination of depth maps, infrared images, RGB, and source-free domain adaptation, with unclear motivation for such integration.
2. The technical approach seems incremental, with widely used GAN-based translation、SHOT and distillation techniques, lacking novel insights or discussions on the suitability of cross-modal representations for source-free domain adaptation.
3. The paper lacks comparisons with recent source-free domain adaptation literature.
4. Writing issues include misalignment of arrows in Table 3.

---

### Official Review · Reviewer_CAob · 2023-11-02

**Soundness:** 3 good
**Presentation:** 3 good
**Contribution:** 2 fair
**Rating:** 3
**Confidence:** 3

**Summary:**

This paper studies source-free cross-modal knowledge transfer, where the key challenge is to adapt models pre-trained with RGB images to other modalities (e.g., depth or infrared) without any source image-label pairs. Under such a setting, the authors propose to make full use of task-irrelevant data (e.g., RGB-depth pairs) to urge the model to learn modality-invariant feature representations. Specifically, two modules are introduced, including Task-irrelevant data-Guided Modality Bridging (TGMB) and Task-irrelevant data-Guided Knowledge Transfer (TGKT). Empirical evidence under various benchmarks demonstrates the effectiveness of the proposed two modules.

**Strengths:**

1. This paper is well-written.
2. The motivation is reasonable.
3. Figure 1 has well-illustrated the motivation and the pipeline.
4. The improvements over baselines are relatively significant.

**Weaknesses:**

**Major concerns:**

1. **The weird setting.** Srouce-Free Domain Adaptation (SFDA) is a similar setting where only a pre-trained model and unlabeled target data are available. However, this setting, i.e., cross-modal knowledge transfer, even assumes that *extra* source-target data pairs are accessible, which is quite weird. If we obtain the predictions of the pre-trained source model over source RGB images, we almost get source labels because the predictions are expected to be relatively accurate. Therefore, the definition of the setting is questionable.

2. **The detailed definition of TR and TI data are unclear,** which makes me confused when going through the paper. Does the TR data refer to source image-label pairs? And does the TI data refer to source-target pairs? A clear definition is strongly encouraged at the beginning of the paper.

3. **Inconsistency in context.** In the related work section, the authors claim that "data generation-based methods may generate noisy source-like images". However, an image translation engine is used in the TGMB module, which makes it inconsistent. How to ensure the adopted image translation model is not noisy?

4. **Missing ablations.** In Table 4, the result of $L_{rec} + L_{IM}$ is missing. In Table 6 and Table 7, the ablations of $\alpha_{im}$ and $\beta_{self}$ are not conducted over a clean baseline (i.e., $\alpha_d$ and $\beta_f$ are not 0 in the table).

**Minor problems/questions:**

5. Is $T(\cdot)$ necessary? What will happen if we simply copy the single-channel depth map into three-channel RGB images?

6. Similar motivation appears in [A], where [A] adopts an image translation engine that transfers source images to the target style to learn domain-invariant feature representations. The source-target image pairs used in [A] are similar to the TI data.

I am looking forward to an open dialog towards my concerns, especially about the setting. I am willing to raise my rating if all of my concerns are well addressed.

**References**

[A] H. Wang et al. Pulling Target to Source: A New Perspective on Domain Adaptive Semantic Segmentation. arXiv preprint arXiv:2305.13752, 2023.

**Questions:**

Please refer to the weaknesses section.

---

### Official Review · Reviewer_CWKx · 2023-11-03

**Soundness:** 2 fair
**Presentation:** 3 good
**Contribution:** 2 fair
**Rating:** 5
**Confidence:** 4

**Summary:**

The paper proposes a novel framework for source-free cross-modal knowledge transfer by leveraging task-irrelevant (TI) data. The authors aim to unlock the potential of paired TI data to estimate the source data distribution and facilitate knowledge transfer from the source model to the target model. The framework consists of two core components: the Task-irrelevant data-guided Modality Bridging (TGMB) module and the Task-irrelevant data-guided Knowledge Transfer (TGKT) module. The TGMB module translates the target modality data into source-like RGB images using domain adversarial learning and the guidance of the source model. The TGKT module facilitates knowledge transfer by minimizing the distance between paired TI data in the feature space. The paper presents extensive experiments on three datasets, demonstrating that the proposed method achieves state-of-the-art performance compared to previous methods. The contributions of the paper include introducing the TGMB and TGKT modules, leveraging task-irrelevant data for knowledge transfer, and addressing the source-free cross-modal knowledge transfer problem.

**Strengths:**

Originality: The paper introduces a novel framework for source-free cross-modal knowledge transfer by leveraging task-irrelevant data. This approach is unique and differentiates itself from existing methods that primarily focus on reducing the modality gap. The use of task-irrelevant data to estimate the source data distribution and facilitate knowledge transfer is an original contribution.

Quality: The paper presents a well-designed framework comprising two core components, the TGMB and TGKT modules, which are supported by technical explanations and experimental evaluations. The proposed method achieves state-of-the-art performance on three benchmark datasets, demonstrating the effectiveness of the approach. The quality of the research is further reinforced by the comprehensive comparisons with baseline methods and the rigorous evaluation conducted.

Clarity: The paper is well-written and provides clear explanations of the proposed framework and its technical components. The authors use concise language and provide visual illustrations to aid understanding. Additionally, the supplementary material provides further details and descriptions to enhance clarity and reproducibility.

Significance: The paper addresses a challenging problem in cross-modal knowledge transfer and presents a solution that has the potential to impact real-world applications. By leveraging task-irrelevant data, the proposed framework offers a source-free approach to estimate the source data distribution and facilitate knowledge transfer between different modalities. The achieved state-of-the-art performance on benchmark datasets further highlights the significance of the research.

Overall, the paper demonstrates originality in its approach, exhibits high quality in its methodology and experimental evaluations, provides clarity in its explanations, and offers a significant contribution to the field of source-free cross-modal knowledge transfer.

**Weaknesses:**

1.	One potential weakness of the paper is the lack of an end-to-end framework. The current approach is non-end-to-end, which may limit its practicality and ease of implementation. Developing an end-to-end framework would enhance the usability and efficiency of the proposed method.
2.	Another weakness is the reliance on the availability of task-irrelevant data. While the paper acknowledges the potential of task-irrelevant data, it does not address the practical challenges of obtaining such data in real-world scenarios. Providing insights or suggestions on how to acquire or utilize task-irrelevant data more effectively would strengthen the applicability of the proposed framework.
3.	Additionally, the paper could benefit from a more thorough discussion of the limitations and potential drawbacks of the proposed method. While some limitations are mentioned, such as the reliance on the source model for guiding the translation process, a more comprehensive analysis of the limitations and their impact on performance would provide a more balanced assessment of the approach.
4.	Lastly, the paper could provide more detailed explanations and justifications for the design choices and hyperparameter settings. This would help readers understand the rationale behind the decisions made and provide insights into potential variations or improvements to the proposed method.
Addressing these weaknesses by developing an end-to-end framework, providing practical solutions for obtaining task-irrelevant data, conducting a more comprehensive analysis of limitations, and offering detailed explanations for design choices would enhance the overall strength and impact of the paper.

**Questions:**

1. Can you provide more details on the process of obtaining task-irrelevant data? How did you select and collect this data, and what considerations were taken to ensure its relevance and usefulness for knowledge transfer?
2. How does the proposed method handle scenarios where the predictions of the task-relevant target data are less reliable? Can you provide insights into the potential impact of unreliable predictions on the overall performance of the framework?
3. Have you considered the potential biases or limitations introduced by using task-irrelevant data for knowledge transfer? How does the presence of task-irrelevant data affect the generalizability and robustness of the proposed method?
4. Can you provide more insights into the trade-off parameter β and its impact on performance? How was this parameter selected, and what considerations were taken to strike a balance between reducing the modality gap and training the model for the task of interest?
5. How does the proposed method compare to existing approaches in terms of computational efficiency? Are there any computational challenges or bottlenecks associated with the proposed framework that could limit its scalability or real-time applicability?
6. Are there any potential ethical or privacy concerns associated with the use of task-irrelevant data? How can these concerns be addressed or mitigated? It is important to consider the ethical implications of using task-irrelevant data for knowledge transfer. The authors should address any potential privacy concerns related to the collection and use of this data. They should also discuss any steps taken to ensure the data is anonymized and does not violate any privacy regulations or guidelines.
7. Can you provide more insights into the limitations of the proposed method? What are the specific scenarios or conditions where the framework may not perform optimally? Are there any known failure cases or challenges that need to be addressed in future work?
8. Have you conducted any user studies or real-world evaluations to assess the practical usability and effectiveness of the proposed method? It would be valuable to understand how the framework performs in real-world scenarios and whether it meets the expectations and requirements of end-users.
9. Have you considered the potential impact of dataset biases on the performance of the proposed method? How does the framework handle biases in the task-irrelevant data and ensure fair and unbiased knowledge transfer?

---

### Official Review · Reviewer_aoSM · 2023-11-04

**Soundness:** 3 good
**Presentation:** 2 fair
**Contribution:** 2 fair
**Rating:** 3
**Confidence:** 4

**Summary:**

In order to transfer the cross-modality knowledge from task-irrelevant data to data that are interested where no source data are available, this paper proposes a method for this knotty problem. The authors first identify that paired task-irrelevant data have the potential to estimate the task-relevant source data distribution, alleviate the modality gap, and facilitate knowledge transfer. Then the authors propose two novel modules for bridging the TI-TR modalities and knowledge transferring. Experiments show that compared with the latest modalities translating methods, the proposed knowledge-transferring module achieves high accuracy in SUN RGB-D dataset, DIML RGB-D, and RGB-NIR datasets with the combination of the proposed modality bridging module.

**Strengths:**

1) reasonable technologies: This paper proposes two modules TGMB and TGKT to bridge the source and target modalities and transfer the knowledge from the source model to the target model. The detailed techniques are sound.

3)Better experimental results: This performance is better than many recent methods in the accuracy of RGB to Depth transfer task and RGB to NIR transfer task, indicating that the proposed modules are effective in bridging the modality gap and facilitating knowledge transfer.

**Weaknesses:**

1) The loss functions in this paper are not well-designed or clearly explained. For example, in equation 1, the RGB reconstruction loss is set as a naïve Mean Square Error loss, while there are some other reconstruction losses more useful that can be used in the RGB reconstruction task. Using the MSE loss alone may cause the reconstructed image not continuous with the adjacent pixels, which may not make the translation net well-trained, and it may further lead to the network not working to its true capacity.

2) The framework in this paper may not convinced to leverage the TI-paired data to their full potential. In the TGMB module, the translation net aims to convert single-channel depth/infrared data into three-channel source-like RGB images, however, it seems too naïve to construct it with an FC layer, a BN layer, and a Conv layer, which may directly cause the failure of the reconstruction process. As Figure 4 in the Appendix shows most images are not well reconstructed.
Still in the TGMB module, the Fs feature extractor’s parameters are fixed all the time, however, it may constrain the generalization ability of the translated TR source data to retrieve their better features. The five classifiers in TGMB and TGKT modules also have the same problem, it is not promised that all these fixed modules will perform to their best ability while not finetuned with the translated new domain data.

3) On some key issues, the paper does not achieve high scores compared with the related works and also lacks an in-depth analysis of the experimental results. First, on SUN RGB-D dataset experiments, the proposed method performs worse than the SOCKET method in K-v2 to K-v1, Real to K-v1, K-v1 to K-v2, Xtion to Real, Xtion to Xtion according to Table 2. It is not proper to say that the proposed method outperforms all other current works while neglecting these lower scores. The only analysis of the performance drop of the K-v1 to K-v2 task just accuses the less reliable mutual information about the TR task for some tasks, and there is no further comprehensive analysis in supplement material about this problem.

**Questions:**

Please refer to the weakness section.
Besides those detailed comments, my major concern lies in the technique's novelty and the overall insights compared to other multi-modal learning frameworks. The authors could give their feedback on this point.

---

### Official Review · Reviewer_QmZs · 2023-11-05

**Soundness:** 2 fair
**Presentation:** 3 good
**Contribution:** 2 fair
**Rating:** 5
**Confidence:** 4

**Summary:**

The paper addresses the novel and practical problem of source-free cross-modal knowledge transfer, and proposes two main modules Task-irrelevant data-Guided Modality Bridging (TGMB) module and Task-irrelevant data-Guided Modality Bridging (TGMB) module to achieve source-free cross-modal knowledge transfer. Extensive experiments show that the method proposed by paper achieves the state-of-the-art performance on three datasets (RGB-to-depth and RGB-to-infrared).

**Strengths:**

The main contribution is a new framework consisting of two key components:
1. Task-irrelevant data-Guided Modality Bridging (TGMB) module, which leverages unlabeled paired data from both modalities to translate the target data into source-like data.
2. Task-irrelevant data-Guided Knowledge Transfer (TGKT) module, which transfers knowledge from the source model to the target model using the task-irrelevant paired data.
3. The authors evaluate the effectiveness proposed method on two cross-modal knowledge transfer tasks with three benchmark datasets: SUN RGB-D, DIML RGB-D), and RGB-NIR. The method achieves a performance improvement of +9.81% on the DIML RGB-D dataset and +3.50% on the RGB-NIR dataset, surpassing the state-of-the-art methods.

**Weaknesses:**

1. Regarding the related work on Cross-Modal Distillation, most of the reviewed papers are 2-3 years old. It would strengthen the literature review to include some more recent works in this area.
2. The experiments are conducted on relatively small datasets with several hundreds to two thousand samples. Does this limit the generalization ability of the method? Please discuss the performance on larger-scale datasets.
3. The paper mentions  that adopting mutual information loss in TGMB can guide the translation process. However, the estimation of mutual information relies on the source model's predictions, which could be unreliable without access to real source data. Did you evaluate the quality of the source model's predictions on the translated data? Please discuss the impact of noisy predictions.
4. The authors just evaluate the proposed method on the classfication tasks, across RGB->Depth/RGB->NIR. Did you validate the approach on other tasks like segmentation or detection? Testing on more diverse tasks would better demonstrate generalizability.
5. While ablation studies validate individual components, the interactions between different losses in TGMB and TGKT are not analyzed. For instance, how does the balance between L_D and L_IM impact translation quality? A more thorough analysis would provide better insights.

**Questions:**

Please check the weakness comments above.